# Effects of Plasma Treatment on the Bioactivity of Alkali-Treated Ceria-Stabilised Zirconia/Alumina Nanocomposite (NANOZR)

**DOI:** 10.3390/ijms21207476

**Published:** 2020-10-10

**Authors:** Seiji Takao, Satoshi Komasa, Akinori Agariguchi, Tetsuji Kusumoto, Giuseppe Pezzotti, Joji Okazaki

**Affiliations:** 1Department of Removable Prosthodontics and Occlusion, Osaka Dental University, 8-1 Kuzuha-hanazono-cho, Hirakata, Osaka 573-1121, Japan; takao-s@cc.osaka-dent.ac.jp (S.T.); akinori@agariguchi.com (A.A.); joji@cc.osaka-dent.ac.jp (J.O.); 2Department of Japan Faculty of Health Sciences, Osaka Dental University, 1-4-4, Makino-honmachi, Hirakata-shi, Osaka 573-1121, Japan; kusumoto@cc.osaka-dent.ac.jp; 3Ceramic Physics Laboratory and Research Institute for Nanoscience, Kyoto Institute of Technology, Sakyo-ku, Matsugasaki, Kyoto 606-8585, Japan; pezzotti@kit.ac.jp

**Keywords:** Ce-stabilised zirconia/alumina nanocomposite, plasma treatment, implant, bone, surface roughness, superhydrophilicity

## Abstract

Zirconia ceramics such as ceria-stabilized zirconia/alumina nanocomposites (nano-ZR) are applied as implant materials due to their excellent mechanical properties. However, surface treatment is required to obtain sufficient biocompatibility. In the present study, we explored the material surface functionalization and assessed the initial adhesion of rat bone marrow mesenchymal stem cells, their osteogenic differentiation, and production of hard tissue, on plasma-treated alkali-modified nano-ZR. Superhydrophilicity was observed on the plasma-treated surface of alkali-treated nano-ZR along with hydroxide formation and reduced surface carbon. A decreased contact angle was also observed as nano-ZR attained an appropriate wettability index. Treated samples showed higher in vitro bovine serum albumin (BSA) adsorption, initial adhesion of bone marrow and endothelial vascular cells, high alkaline phosphatase activity, and increased expression of bone differentiation-related factors. Furthermore, the in vivo performance of treated nano-ZR was evaluated by implantation in the femur of male Sprague–Dawley rats. The results showed that the amount of bone formed after the plasma treatment of alkali-modified nano-ZR was higher than that of untreated nano-ZR. Thus, induction of superhydrophilicity in nano-ZR via atmospheric pressure plasma treatment affects bone marrow and vascular cell adhesion and promotes bone formation without altering other surface properties.

## 1. Introduction

In the field of dentistry, yttria-stable zirconia (YSZ) and ceria-stabilized zirconia/alumina nanocomposites (nano-ZR) show superior mechanical properties, biocompatibility, and aesthetic performance. Zirconia-based materials are also superior to titanium implants in terms of discoloration and responsiveness [1,2]. At the tissue level, the biocompatibility of zirconia is comparable to that of titanium [2,3,4]. Rough zirconia implants have higher removable torque values than control implants, but not at the levels observed with titanium implants [5,6,7,8,9]. Therefore, various surface-roughening techniques have been investigated to enhance the biological activity and osseointegration capabilities of titanium, including physical approaches such as compaction of nanoparticles [10], ion beam deposition [11]; chemical methods such as acid etching, peroxidation [12], and anodization [13]; nanoparticle deposition such as discrete crystalline deposition [14] and lithography and contact printing technique [15]; however, little is known about the clinical effects of zirconia implants. As the speed and quality of osseointegration are closely related to the implant surface properties, titanium and zirconia implant surfaces meet the optimal response and high biocompatibility requirements of the adjacent osteogenic cell population [16,17,18]. The cell/tissue development, chemical composition, energy, topography, and roughness at the tissue–biomaterial interface are the most studied surface properties for regulation of bone formation [18,19,20].

Nano-ZR is a nanocomposite in which nanometer-sized alumina particles and ceria-stabilized zirconia particles are dispersed in ceria-stabilized zirconia and alumina crystals, respectively. It has flexural strength and fracture toughness higher than the respective corresponding parameters of yttria-type zirconia materials widely used in dentistry, as described by Nawa et al. [21,22]. Additionally, compared to 3Y-TZP, nano-ZR is resistant to the low-temperature degradation that affects zirconia [23,24,25]. The fatigue strength of nano-ZR, as measured by the cyclic test, is twice than that of 3Y-TZP [21,22].

Li et al. have reported that the apatite layer formed on the surface of titanium and titanium alloys by alkali treatment is involved in osseointegration [26]. It has also been reported that acid treatment results in changes in the surface structure of titanium and zirconia surfaces, and plays a role in rat bone marrow cell behavior and hard-tissue formation [27,28,29]. In previous studies, the concentrated alkali treatment of titanium and titanium alloys was shown to induce hard-tissue differentiation [30,31,32,33,34,35,36]. Nanostructures formed on the material surface affected both protein adsorption and cell behavior, thereby indicating that alteration of the surface structure could be advantageous for osteointegration. We have shown that alkali-modified nano-ZR can accelerate osseointegration both in vitro and in vivo. Alkali-modified nano-ZR helps improve early adhesion and differentiation of rat bone marrow mesenchymal stem cells compared with unmodified nano-ZR surfaces [37,38]. These results suggest that alkali-modified nano-ZR is useful as a novel implant material, similar to the previously reported use of alkali-modified titanium [37]. Our latest study has also shown that alkali-modified nano-ZR treatment is effective for peri-implant hard-tissue formation [37,38]. However, compared to the biocompatibility of titanium surfaces with controlled nanostructures, the biocompatibility of alkali-treated nano-ZR is insufficient, and therefore development of new surface treatments is needed.

Plasma biology is a new interdisciplinary research area [39]. The range of uses of plasma processing continues to expand, and it is being applied to various fields. Currently, functionalization of the biological material surface to improve biocompatibility is being used to strengthen biologically equivalent coatings [40,41,42]. The relationship between plasma treatment of implant material surfaces and hard-tissue formation has been reported in several studies [43,44,45,46]. In the past few decades, several methods have been used to generate plasma at near-atmospheric and ambient temperatures, such as radio frequency plasma [47], dielectric barrier discharge plasma [48], corona discharge plasma [49], and glide arc discharge plasma [50]. The general advantage of these techniques is that they allow the formation of a large number of reactive species used for surface, gas, and aqueous treatments. Owing to the large size of the plasma device that is applied clinically, downsizing of the plasma generator is an important goal. The purpose of these technologies, such as piezo direct discharge plasma, is to produce as thin and small a plasma device as possible from viewpoint of clinical applications. However, devices that perform plasma processing are very large in size and are not practical for clinical applications. In contrast, the plasma device used in the present study is relatively small, easy to use, and very useful in clinical settings. We have already reported that atmospheric pressure plasma treatment of titanium implants increases the hydrophilicity of the material surface and affects the initial adhesion of rat bone marrow mesenchymal stem cells (rBMMSCs) as well as the ability to induce hard-tissue differentiation. Therefore, this procedure may also be applied to other materials such as zirconia [51].

The present study aimed to investigate the mechanisms by which piezobrush plasma treatment applied to alkali-modified nano-ZR surface affected the initial adhesion of rat bone marrow mesenchymal stem cells, their ability to induce hard-tissue differentiation, and their efficacy in promoting hard-tissue formation in the implant surroundings. The results of the present study will provide valuable and novel insights into the fields of dentistry and prosthodontics.

## 2. Results

### 2.1. Evaluation of Nano-ZR Samples

The evaluation of nano-ZR samples is depicted in Figure 1. No change was observed in the mechanical structure of the material surface by scanning electron microscopy (SEM) analysis and scanning probe microscopy (SPM) analysis. X-ray photoelectron spectroscopy (XPS) analysis showed an increase in the O1s peak and a decrease in the C1s peak on the material surface of the test group. (Figure 1). Plasma treatment showed a remarkable decrease in contact angle on the surface of the material, resulting in superhydrophilicity in the test group (Figure 1).

### 2.2. Evaluation of Protein Adsorption on the NANO-ZR Surface

The adsorbed amount of bovine serum albumin (BSA) on the material surface of the test group was significantly higher than that of the control group at all measurement times (Figure 2).

### 2.3. Effects of the Nano-ZR Surface on Cell Adhesion and Morphology in rBMMSCs and HUVECs

The morphology of rat bone marrow mesenchymal stem cells (rBMMSCs) and human umbilical vein endothelial cells (HUVECs) on the nano-ZR surface after 24 h of culture was observed with a fluorescence microscope. It was confirmed that various cells adhered to the surface of the materials in both groups (Figure 3). Simultaneously, an increase in the number of cells and elongation of cell projections was observed on the material surface of the test group compared with that of the control group. In this experiment, cell morphology was observed and the number of cells on the surface of each material was compared. At all measurement times, the adhesion number of rBMMSCs and HUVECs in the test group was significantly higher than that in the control group.

### 2.4. Evaluation of Hard Tissue Differentiation and Angiogenesis of rBMMSCs and HUVECs on Nano-ZR In Vitro

The gene expression related to the induction of hard tissue differentiation and angiogenesis on the material surface of the test and the control group was analyzed. In this experiment, the assay was performed at a measurement time specific to each gene. Significantly higher gene expression was observed on the material surface of the test group at all measurement times (Figure 4a,b). Alkaline phosphatase (ALP) expression was measured for differentiation as the initial marker for the induction of hard tissue differentiation. ALP expression in bone marrow cells at 7 d and 14 d after the start of culture was significantly higher on the material surface of the test group (Figure 4c). Mineralization was assayed for calcification, which is a late marker for the induction of hard tissue differentiation. The amount of Ca deposited at 21 d and 28 d after the culture incubation was significantly high on the material surface of the test group (Figure 4d).

### 2.5. Evaluation of the Amount of New Bone Formation in the Tissue Surrounding the Nano-ZR Implant Placement In Vivo

More trabecular microarchitecture was observed in the area of the material surface in the test group than in the area of the control group (Figure 5). Furthermore, the ratio of bone mass to total mass (BV/TV), average trabecular number (Tb.N), and average trabecular thickness (Tb.Th) were significantly higher in plasma TNS samples and plasma treated samples. Implants promoted osteogenic activity (*p* < 0.05). Conversely, mean trabecular separation (Tb.Sp) was significantly lower in the test group than that in the control group.

Furthermore, the amount of new bone formation was confirmed using longitudinal sections. As shown in Figure 6, more newly formed bone was observed around the implants in the test group than that in the control group. Quantitatively, histomorphometric analysis showed that bone area ratio (BA) and bone-to-implant contact (BIC) were significantly higher around the test implants than those around the control implants (Figure 6). Additionally, newly formed bone around the implant was favorably labeled with oxytetracycline hydrochloride (blue) at 1 w, with alizarin red S (red) at 4 weeks, and with calcein (green) at 8 weeks. The labeled bone area between the implant interface and the labeled bone area at 1, 4, and 8 weeks was significantly higher in the test group than that in the control group (*p* < 0.05; Figure 7).

## 3. Discussion

The surface of dental restorations can be modified by plasma treatment. Increased surface energy improves the adhesive bond between the restorative material and the tooth and between different restorative materials [52,53,54]. This method utilizes the phenomenon of ionizing electrons and cations when a high voltage is applied to air, mainly oxygen, and is often used for surface modification [55,56]. The accelerated electrons break the interatomic bonds of oxygen atoms and produce active oxygen radicals. Ozone is produced in the plasma, and an oxidative decomposition reaction occurs owing to the active oxygen species generated therefrom [57,58,59]. The mechanism involves decomposition of the hydrocarbons adsorbed onto the surface by the generated active oxygen. This implies that the mechanical change on the material surface is caused by a chemical change rather than by plasma treatment. As shown by SEM and SPM analysis, no mechanical change occurred on the material surface. Additionally, XPS results showed that a significant decrease in carbon occurred first. Carbon, is a hydrophobic organic substance on the surface of a material, is known to inhibit the initial adhesion of bone marrow cells. Therefore, reduction in carbon by atmospheric pressure plasma treatment is useful for hard-tissue formation in the implant surroundings. XPS analysis shows that the composition ratio of zirconium and oxygen increased. This indicates that oxygen is activated by the plasma treatment, and that the material surface is in an oxygen-rich state [60,61]. Measurement of the contact angle revealed that the material surface exhibited superhydrophilicity resulting from the atmospheric pressure plasma treatment. It is reported that hydrophilization of the material surface correlates with the suppression of inflammatory cytokine accumulation, enhancement of adhesion in cell and bone proteins, promotion of osseointegration and soft-tissue adhesion, and suppression of biofilm formation [62,63]. The results of the present study are consistent with those obtained in prior studies, and indicate that atmospheric pressure plasma treatment of nano-ZR is useful for clinical applications. Previous studies have shown that there is not always a correlation between a decrease in carbon content on a material surface and a decrease in the contact angle [64]. In our previous study, XPS analysis revealed that it was possible to form hydroxyl groups on the nano-ZR surface with concentrated alkali treatment [37,38]. The present study shows that the combination of carbon decomposition on the material surface by atmospheric pressure plasma treatment imparts superhydrophilicity to the material surface. When titanium or zirconia is subjected to plasma treatment, organic substances are decomposed by collision of high-energy molecules. In addition, the oxide is excited by oxygen in the plasma state, and at the same time, water molecules in the environment react to form a hydroxyl group. It is presumed that the carbon decomposition effect on the nano-ZR surface and the introduction of hydroxyl groups exerted superhydrophilicity. Our previous study showed that treatment of nano-ZR with a concentrated alkali solution changes the zeta potential of the material surface from negative to positive [38]. Choi et al. also revealed that application of atmospheric pressure plasma treatment to the implant material surface could render the zeta potential of the material surface positive [64]. In line with these results, the zeta potential of the material surface in this experiment is expected to be positive and the surface is expected to be involved in the adsorption of proteins related to cell adhesion. In Figure 8, we have suggested the possible effects of alkali treatment and plasma treatment on nano-ZR surfaces. 

In the in vitro evaluation, we investigated the initial reaction of bone marrow cells and vascular endothelial cells to the material surface immediately after implantation, based on the results of our research. Hydrophilic surface treatments such as plasma treatment enhance the adhesion of BSA, vascular endothelial cells, and rBMMSCs. In cell-substance interactions, protein adsorption is one of the first events to occur between the implant material and the body when the implant is processed during implantation [65,66,67,68,69]. These results indicate that changes in surface structure are linked with initial adhesion and proliferation of various cells. Observation of the response of HUVECs to the material surface after implant surgery is important in wound healing considerations. We found that plasma treatment was useful for the initial adhesion of HUVECs on the surface of nano-ZR material and the enhancement of gene expression related to angiogenesis. Further, use of plasma-treated materials contributes to accelerated wound healing after implant placement. The decrease in the contact angle of the material surface and the chemical change of the material surface by plasma treatment also affect the behavior of cells on various material surfaces. Such changes in the material surface also have a large effect on the markers, thereby indicating the induction of hard tissue differentiation.

The present study revealed that rBMMSCs cultured on nano-ZR displayed a clear ability to induce hard-tissue differentiation. The plasma treatment imparted superhydrophilicity and reduced carbon on the surface of the nano-ZR treated with alkali, which correlated with the ability to induce angiogenesis, initial adhesion of bone marrow cells, and hard-tissue differentiation. In our study, nano-ZR with superhydrophilicity resulting from alkali and plasma treatment promoted RBM cell adhesion and osteogenic differentiation, and may effectively promote bone integration into implants.

In vivo evaluation of material surfaces is essential for biomaterial development. We recently reported related results using an in vivo model with Sprague–Dawley rat femurs [68,69,70,71]. The rat femur model in the present study was used to evaluate the bone tissue response to implants via inclusion of the trabecular bone in the clinical context. In the rat model, healing at this point has been considered to be in the final stages, but the degree of BIC was affected by the initial bone response to the implant surface. It is known that the bone response to the surface of zirconia materials is slightly slower than the response to titanium. In our study, no clear bone formation was observed at 1 w after implantation whereas, at 4 weeks, a clear difference in bone formation was observed between the test group and the control group. At w 8, bone formation was almost complete in the plasma-treated group; thus, we hypothesized that hard-tissue formation would be almost the same as that in alkali-treated titanium metal surfaces shown in our previous study. The piezobrush used in these experiments is a low-temperature, atmospheric-pressure plasma treatment method that creates an ionized state at low temperatures. Compared to UV treatment, the plasma treatment time is short, although there is a concern that its effects too may be short-lived. However, our results clearly show that the initial response of the material surface and the implanted tissue after plasma treatment allow rapid progress from angiogenesis to hard-tissue formation. Therefore, based on our results, we propose that this material may be useful for patients with metal allergies.

## 4. Materials and Methods 

### 4.1. Sample Preparation

Nano-ZR discs (15 mm in diameter and 1.5 mm in thickness; Yamamoto Kinzoku, Osaka, Japan) and screw implants (1.2 mm in external diameter and 12 mm in length) were used in the present study. Samples were soaked in 10 M aqueous NaOH for 24 h at 30 °C. The solution in samples was replaced with distilled water (200 mL) until the conductivity reached 5 μS/cm and then dried at room temperature. Plasma treatment was performed using a non-thermal atmospheric pressure handheld plasma device (Piezobrush^®^ PZ2, Relyon Plasma GmbH, Regensburg, Germany) that utilized piezoelectric direct discharge technology. Half of the samples were treated at room temperature with plasma induced by active gas at atmospheric pressure for 30 s. The distance between the jet exit and samples was set to 5 mm to ensure that the samples were wholly immersed in the plasma plume emerging from the nozzle. Plasma-treated nano-ZR was tested in the test group, while the control group was untreated.

### 4.2. Characterization of NANO-ZR Samples

Scanning electron microscopy (SEM) (S-400; Shimadzu, Kyoto, Japan) and scanning probe microscopy (SPM) (SPM-9600; Shimadzu) were used for comparative evaluation of the surface properties of alkali-modified plasma-treated and untreated nano-ZR discs. X-ray photoelectron spectroscopy (XPS) (Kratos Analytical Axis Ultra DLD electron spectrometer; Kratos Instruments, Manchester, UK) with a monochromatic Al Kα X-ray source was used to analyze the components of samples after argon-ion etching for 2 min (evaporation rate 5 nm/min). This procedure was performed to remove surface contaminants. Sample surface wettability was analyzed using a contact angle measurement system (VSA 2500 XE, AST Products, Tokyo, Japan). The material used was ultrapure water.

### 4.3. Protein Adsorption

Pierce^TM^ BCA Protein Assay Reagent Kit (Pierce Biotechnology) was used by evaluating protein adsorption on nano-ZR disks of test and control group. BSA fraction V solution (1 mg/mL protein in saline; Pierce Biotechnology, Rockford, IL, USA) was used as samples, A volume of 300 µL of was placed onto both test and control discs. The evaluation time was 1, 3, 6, and 24 h after dropping BSA on samples. After the lapse of various times, BSA not adhered to the nano-ZR surface was recovered. The amount of adhered BSA was measured from the amount of BSA not adhered.

### 4.4. Cell Culture 

RBMMSCs were obtained from the femurs of 8-w-old Sprague-Dawley rats (SHIMIZU Laboratory Supplies Co., Kyoto, Japan). The hind limb bones of the rats were aseptically excised after euthanizing them with 4% isoflurane. The proximal of the mesial and distal ends of the rats’ tibia were clipped. The marrow was flushed from the shaft with culture solution by inserting a 21-gauge needle (Terumo, Tokyo, Japan) into the hole in the knee joint of each bone. Cell suspensions from all bones were combined in a centrifuge tube and then the resultant marrow pellet was spread by trituration. The animal experiment was performed according to the ethical principles of the National Animal Care Guidelines and was approved by the Medical Ethics Committee of Osaka Dental University, Japan (approval no. 19-06001; August 16, 2019). 

Human umbilical vein endothelial cells (HUVECs) were purchased from CellWorks (Buckingham, UK). The culture was performed on a type 1 collagen-coated disc according to the culture method that we have already reported [35].

RBMMSCs and HUVECs were removed from each flask at 3rd–5th culture, and seeded at a cell density of 4 × 10^4^ cells/well into 24-well tissue culture plates (BD Biosciences, Franklin Lakes, NJ, USA) including test or control nano-ZR disks. 

### 4.5. Cell Adhesion

RBMMSCs and HUVECs were seeded onto the specimens at an initial density of 4 × 10^4^ cells/cm^2^ and allowed to attach for 1, 3, 6 and 24h. The number of rBMMSCs and HUVECs adhesions was examined by CellTiter-Blue^®^ Reagent (Promega, Madison, WI, USA, 50 μL CellTiter-Blue^®^ Reagent diluted in 250 μL PBS). The analysis method follows the manufacturer’s instructions.

According to our past report, various cells after 24 h of culture were stained and observed [31,32,33,34,35,36,37,38].

### 4.6. QRT-PCR, Alkaline Phosphatase Activity, DNA Content, and Calcium Deposition

Expression of genes was assessed using a real-time TaqMan RT-PCR assay (Life Technologies, Carlsbad, CA, USA). Total RNA was extracted using an RNeasy Mini Kit (Qiagen, Venlo, the Netherlands), and 10-μL aliquots of each RNA sample were reverse transcribed into cDNA utilizing a Prime Script RT Reagent kit (Takara Bio, Shiga, Japan). We investigated *alkaline phosphatase (ALP)* on d 7, *runt-related transcription factor (Runx2)* on d 3, *bone morphogenetic protein 2 (Bmp-2)* on d 14, *Bglap* on d 21 of RBMCs. We also investigated *ICAM-1* on d 3, *von Willebrand factor* on d 7, and *thrombomodulin* mRNAs on d 14 of HUVECs.

In order to evaluate ALP activity, following 7 or 14 ds of incubation, samples were washed with PBS, and cells that had attached to the sample surface were dissolved with 300 μL of 0.2% Triton X-100. ALP activity was evaluated by an alkaline phosphatase luminometric enzyme-linked immunosorbent assay (ELISA) kit (Sigma-Aldrich, St. Louis, MO, USA) in accordance with the manufacturer’s instructions. A PicoGreen dsDNA analysis kit (Invitrogen/Life Technologies) was utilized to evaluate the DNA content. The amount of ALP was normalized to the amount of DNA in each cell lysate.

Following 21 or 28 ds of incubation, calcium deposition in the extracellular matrix was measured after dissolution with 10% formic acid. Calcium content was quantified and calculated using a Calcium E-test Kit (Wako Pure Chemical Industries, Osaka, Japan) according to the manufacturer’s instructions.

According to our past report, PCR analysis, ALP activity, and calcium deposition were analysed [31,32,33,34,35,36,37].

### 4.7. Animal Model and Surgical Procedures

8-week-old male Sprague-Dawley rats (Shimizu Laboratory Supplies Co., Kyoto, Japan) were used in this study. The rats were randomly divided into two groups, with eight rats in each group. Surgical procedures used in this study were previously described [38]. After general anesthesia and surgical cleaning, a 10 mm longitudinal incision was made along the medial side of the knee joint of the right hind leg. The patella and extensor mechanism were then dislocated to expose the distal femur. A 1.2 mm hole was drilled into the intercondylar notch using a dental burr with sterilized saline irrigation. Screws were implanted into the prepared channels, the knee joint was restored, and the incision was sutured. Gentamicin (1 mg/kg) and buprenorphine (0.05 mg/kg) were injected for 3 d after surgery to prevent post-surgical infection and decrease postoperative pain.

According to our past report, surgical procedures of animal models were used [38].

### 4.8. Sequential Fluorescent Labeling

Polychrome sequential labeling of bone via intraperitoneal injection of fluorescent dyes was employed to determine the process and characteristics of new bone formation and mineralization after implantation according to the following timetable: rats were injected with 25 mg/kg oxytetracycline hydrochloride (Sigma-Aldrich, USA) at 1 week after implantation, with 30 mg/kg alizarin red S (011-01192, Wako, Japan) at 4 weeks, and with 20 mg/kg calcein (340-00433, Wako, Japan) at 8 weeks. 

According to our past report, sequential fluorescent labelling of in vivo model was used [38].

### 4.9. Morphological Analysis

Rats were then anesthetized and euthanized at 8 weeks, and the right femurs including the implants were placed in a saline solution immediately after dissection and scanned with an SMX-130CT microcomputed tomography (micro-CT) scanner (Shimadzu) operated at 70 kV and 118 mA. Three-dimensional reconstruction models were obtained using morphometric software (TRI/3D-BON; Ratoc System Engineering, Tokyo, Japan). The region of interest was defined as 2 mm below the highest point of the growth plate and extending 500 μm around each implant. The bone volume fraction (BV/TV), mean trabecular number (Tb.N), mean trabecular thickness (Tb.Th), and mean trabecular separation (Tb.Sp) were quantified to assess bone regeneration.

After the micro-CT scan, implanted femurs collected at 8 weeks were stained utilizing the Villanueva method to evaluate bone generation. All histomorphometric and fluorescence characteristics of the sections were analyzed using a BZ-9000 digital cold illumination microscope (Keyence Co., Osaka, Japan) and a laser scanning microscope (Carl Zeiss, Oberkochen, Germany), respectively. The excitation and emission wavelengths were 351/460 nm for oxytetracycline hydrochloride (blue), 543/617 nm for alizarin red S (red), and 488/517 nm for calcein (green), respectively. Bone area, BIC, and labeled bone area were assessed utilizing ImageJ software (TRI/3D-BON; Ratoc System Engineering, Tokyo, Japan) in a 200× field around the implant.

### 4.10. Statistical Analyses

All data were expressed as the mean ± standard deviation. Each experiment was repeated three times, and all results were compared in SPSS 26.0 by the Student’s *t*-test; *p* < 0.05 was considered statistically significant.

## 5. Conclusions

The results of the present study demonstrated that plasma treatment applied to alkali-modified nano-ZR surfaces increased the biocompatibility nano-ZR -based materials. The surface analysis of the concentrated alkali-treated nano-ZR treated with atmospheric pressure plasma showed no difference in the mechanical structure of the surface, but showed a decrease in carbon on the surface, indicating super hydrophilicity. The treatment enhanced the angiogenic induction and osteogenic induction abilities of alkali-modified nano-ZR, as well as the adherence of HUVECs and rBMMSCs. In addition, when the plasma-treated alkali-modified nano-ZR screw was implanted into a rat femur, compared to the untreated group, we showed that more new bone was formed in the tissue surrounding the implant. Therefore, based on both in vitro and in vivo results, we conclude that the atmospheric pressure plasma-treated nano-ZR material subjected to a concentrated alkali treatment and atmospheric pressure plasma treatment is useful as a prosthetic treatment option for patients allergic to metal. 

## Figures and Tables

**Figure 1 ijms-21-07476-f001:**
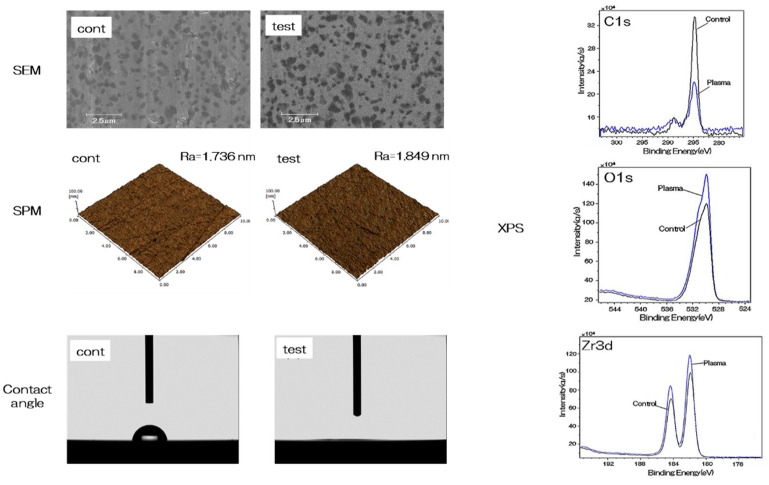
The surface analysis of the test and control NANOZR samples were showed. SEM analysis showed images specific to NANOZR in both the test group and the control group, and no change due to plasma treatment was observed. In addition, the SPM analysis showed Ra of 1.736 nm in the control group and 1.840 nm in the experimental group, showing no difference in surface roughness. In XPS analysis (C1s, O1s and Zr3d), decrease of C1s peaks and increase of O1s and Zr3d peaks were observed on the material surface of the test group by alkali and plasma treatment to NANOZR surface. The contact angle analysis showed 63° on the material surface of the control group, while showing superhydrophilic on the material surface of the test group.

**Figure 2 ijms-21-07476-f002:**
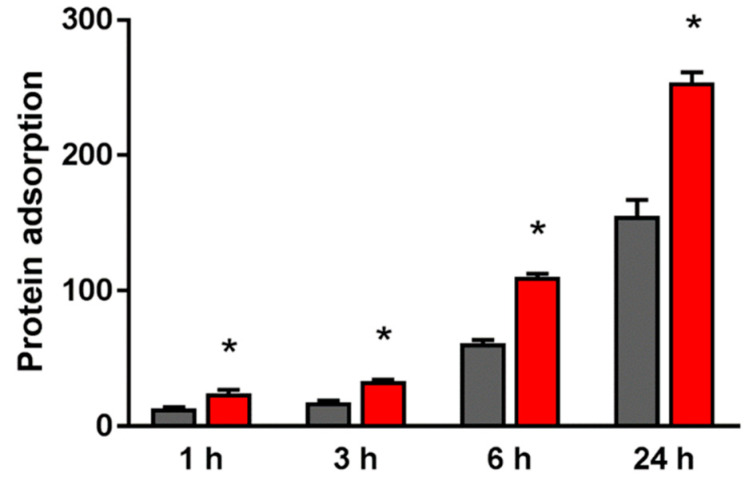
The adsorption amount of BSA on the material surface of the test group (red) and that of the control group (black). * *p* < 0.05.

**Figure 3 ijms-21-07476-f003:**
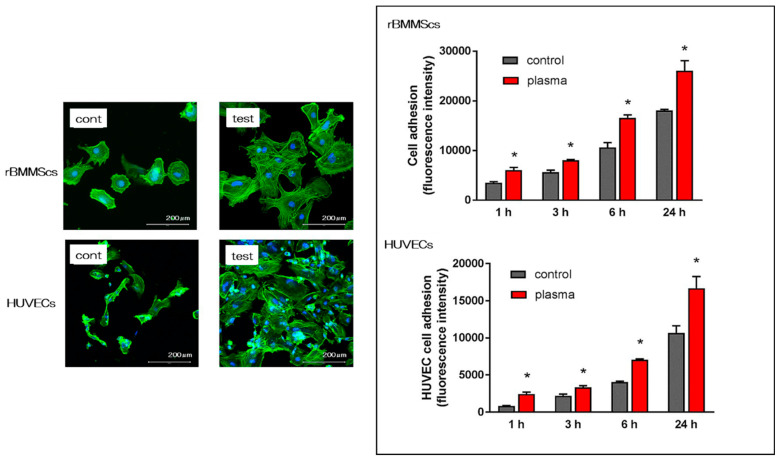
As a result of observation after 24 h of culture with a fluorescence microscope, images showing cell adhesion of rBMMSCs and HUVECs were observed on the material surface in both groups. By alkali and plasma treatments, elongation of cell processes of both cells was observed. In addition, when the number of initial rBMMSCs adhesion and HUVECs was compared at each culture time, it was found that the number of cell adhesion increased with the passage of culture time. Furthermore, the material surface of the test group showed a statistically significantly higher value than that of the control group at all measurement times. * *p* < 0.05.

**Figure 4 ijms-21-07476-f004:**
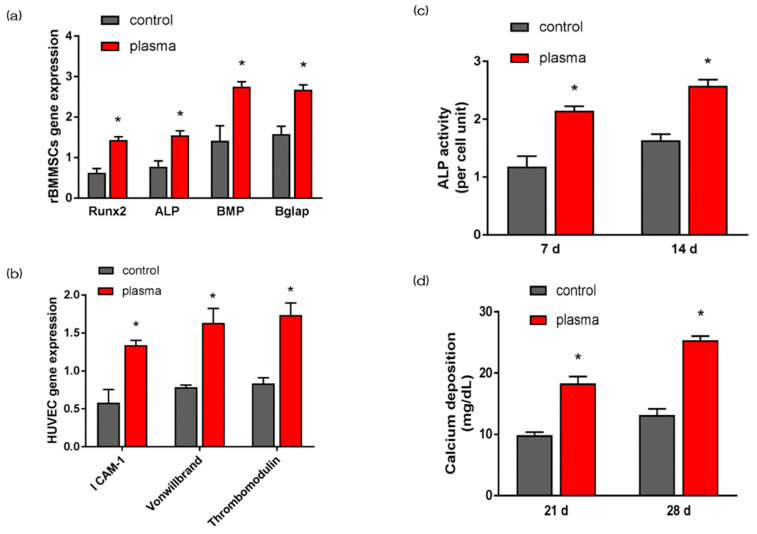
Quantitative real-time (qRT)-PCR analysis of osteogenesis and angiogenesis related gene expression in control and test groups (**a**,**b**). ALP activity in control and test groups (**c**). Ca deposition in control and test groups (**d**). * *p* < 0.05.

**Figure 5 ijms-21-07476-f005:**
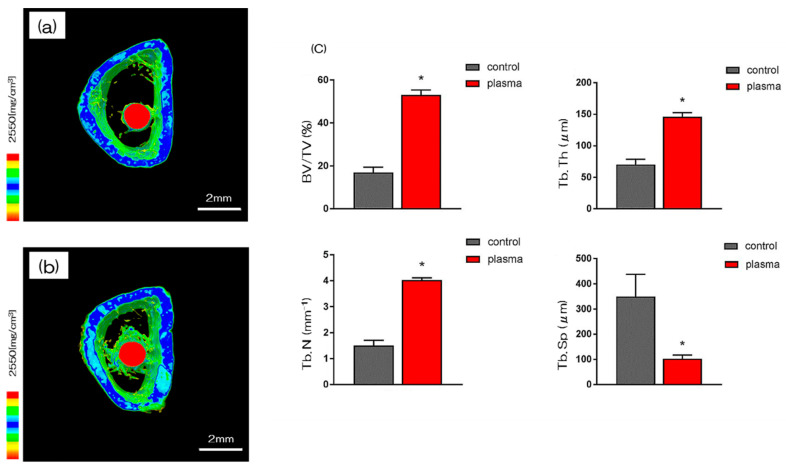
The reconstructed three-dimensional macro-CT images of rat femurs transverse slices showed that thin new bone layers are formed round the implants in the test and control group after 8 weeks. (**a**; control, **b**; test) In both (**a**,**b**), new bone formation was observed around the implant model, but in (**b**), more new bone formation was observed. (**c**) The BV/TV, Tb.N, and Tb.Th were significantly higher in the test group samples than that of the control group (* *p* < 0.05). Conversely, Tb.Sp was significantly lower in the test group than that of the control group.

**Figure 6 ijms-21-07476-f006:**
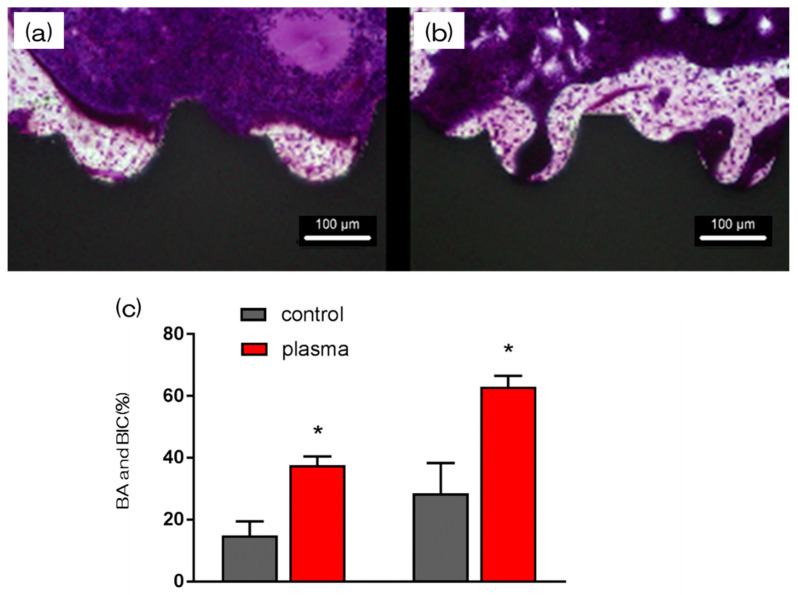
The histological sections showed that control implants still exhibited fibrous connective tissue from the bone-implant interface, whereas the test implants were almost entirely surrounded with new bone (**a**; control, **b**; test). The part indicated by the red arrow is the new-formed bone. In both (**a**,**b**), new bone formation was observed around the implant model, but in (**b**), more new bone formation was observed. (**c**) Furthermore, histomorphometric analysis showed that bone area ratio (BA) and bone-to-implant contact (BIC) were significantly higher around the test implants than around the control implants (* *p* < 0.05).

**Figure 7 ijms-21-07476-f007:**
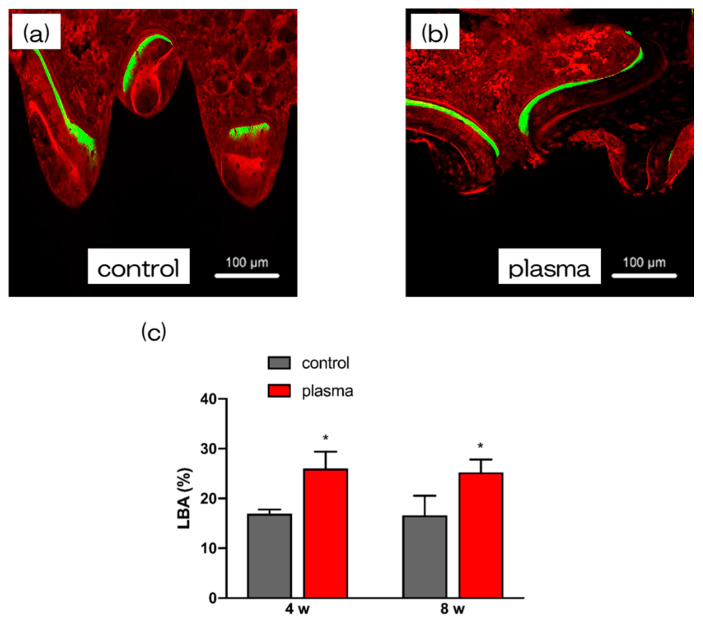
The sequential fluorescent labeling was also performed by scanning microscopy to analysis the process of bone formation. (**c**) The test implants showed increased fluorescent areas than control implants, indicating that alkali-plasma treatment might endow titanium with better bone deposition and remodeling at each time point. (* *p* < 0.05). It is clear from (**a**,**b**) that the dyes of the two colors of 4w and 8w were dyed, although the dyes of three colors were used in this experiment. However, when both stained images were confirmed, it was clear that the amount of new bone formation in group b was high in (**a**,**b**).

**Figure 8 ijms-21-07476-f008:**
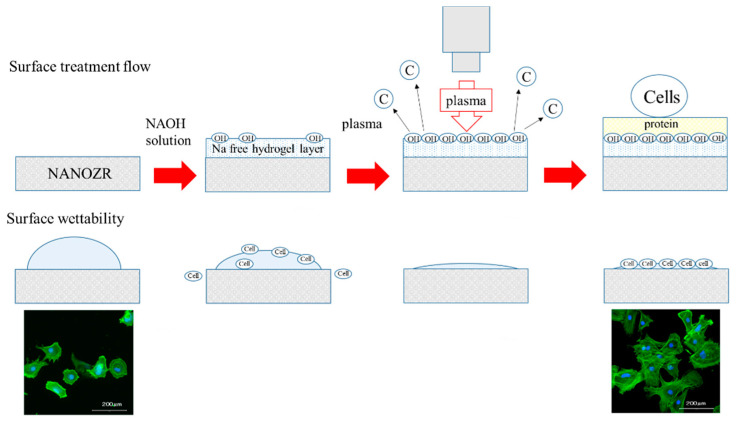
When the nano-ZR surface is subjected to alkali treatment, a hydrogel layer of Na ions is formed. Upon plasma treatment, formation of hydroxyl groups and removal of carbon are observed on the surface. After that, protein is adsorbed to this layer, and the cells are attached using the protein for binding.

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
