# Peer review of "Effects of Plasma Treatment on the Bioactivity of Alkali-Treated Ceria-Stabilised Zirconia/Alumina Nanocomposite (NANOZR)"

_ijms, 2020, doi:10.3390/ijms21207476_

Round 1
Reviewer 1 Report
The reviewer has serious concerns about the manuscript authored by Takao et al. with the title “Effects of plasma treatment on the bioactivity of alkali-treated ceria-stabilized zirconia/alumina nanocomposite (NANOZR)” as detailed below.
Cell culture studies: cell isolation should be briefly described. The passage of the cells was not mentioned.
How were the cells characterized before seeding?
Bone mineral density measurements are missing, should be performed.
P10L332: Reference is missing.
The majority of the references are old, the state of the art and the key papers should be cited.
The mentioned supplementary file is missing.
The first sentence of the abstract should be something else than the current one.
All upper case abbreviations should be avoided if possible such as NANOZR (nano-Zr reads much better than NANOZR).
Unnecessary coloring of the graphs should be avoided, if colors do not have any specific meaning then b/w or grayscale should be preferred.
The use of abbreviations/long forms is not uniform (such as it should either “day” or “d” not a mixture).
Scale legends in Fig. 5 cannot be read.
Author Response
Thank you very much for circulating our manuscript entitled “Effects of plasma treatment on the bioactivity of alkali-treated ceria-stabilised zirconia/alumina nanocomposite (NANOZR)” by Seiji Takao et al. among the members of the editorial board of the International Journal of Molecular Scineces and forwarding two reviewers’ suggestions to me. I have enclosed our responses to the reviewers’ comments.
Reviewer #1:
Thank you very much for your comments. We have revised our manuscript in accordance with your suggestions as follows:
- Cell culture studies: cell isolation should be briefly described. The passage of the cells was not mentioned.
We agree with your suggestion and have corrected our manuscript as follows:
Page 10, lines 300-314:
RBMCs were obtained from the femurs of 8-week-old Sprague-Dawley rats (SHIMIZU Laboratory Supplies Co., Kyoto, Japan). The hind limb bones of the rats were aseptically excised after euthanising them with 4% isoflurane. The proximal of the mesial and distal ends of the rats’ tibia were clipped. The marrow was flushed from the shaft with culture solution by inserting a 21-gauge needle (Terumo, Tokyo, Japan) into the hole in the knee joint of each bone. Cell suspensions from all bones were combined in a centrifuge tube and then the resultant marrow pellet was spread by trituration. The animal experiment was performed according to the ethical principles of the National Animal Care Guidelines and was approved by the Medical Ethics Committee of Osaka Dental University, Japan (approval no. 19-06001; August 16, 2019).
Human umbilical vein endothelial cells (HUVECs) were purchased from CellWorks (Buckingham, UK). The culture was performed on a type 1 collagen-coated disc according to the previously reported culture method [30].
RBMCs and HUVECs were removed from respective flasks at 3rd-5th culture, and seeded at a density of 4 × 104 cells/well into 24-well tissue culture plates (Falcon) containing test or control NANOZR disks.
- How were the cells characterized before seeding?
We already characterized RBMCs in recent articles. Besides, HUVECs were characterized by Cellworks.
- Nishizaki, M.; Komasa, S.; Taguchi, Y.; Nishizaki, H.; Okazaki, J. Bioactivity of NANOZR induced by alkali treatment. Int J Mol Sci 2017, 18, 780.
- Komasa, S.; Nishizaki, M.; Zhang, H.; Takao, S.; Kobayashi, Y.; Kusumoto, T.; Yoshimine, S.; Nishizki, H.; Okazaki, J.; Chen, L. Osseointegration of alkali-modified NANOZR implants: an in vivo study. Int J Mol Sci. 2019, 20, 842.
- Komasa, S.; Nishizaki, M.; Kusumoto, T.; Terada, C.; Derong, Y.; Kawamoto, A.; Yamamoto, S.; Yoshimine, S.; Nishizaki, H.; Shimizu, H.; Okazaki, J.; Kawazoe, T. Osteogenesis-related gene expression on alkali-modified NANOZR and titanium surfaces with nanonetwork structures. J Bio-integration 2017, 7, 87–94.
- Bone mineral density measurements are missing, should be performed.
Thank you for your suggestion. Bone density is not evaluated in this paper. Therefore, the titles of 4. 6 have been modified as follows.
Page 10, lines 324:
4.6 qRT-PCR, alkaline phosphatase activity, DNA content, and calcium deposition
- P10L332: Reference is missing.
Thank you for your suggestion. We added references.
- The majority of the references are old, the state of the art and the key papers should be cited.
We agree with your suggestion and have corrected the reference.
- The first sentence of the abstract should be something else than the current one.
We agree with your suggestion and have corrected our manuscript as follows:
Page 1, lines 16-18:
Zirconia ceramics such as ceria-stabilized zirconia/alumina nanocomposites (NANOZRs) are applied as implant materials due to their excellent mechanical properties. However, surface treatment is required to obtain sufficient biocompatibility.
- All upper case abbreviations should be avoided if possible such as NANOZR (nano-Zr reads much better than NANOZR).
Thank you for your suggestion. We changed nano-Zr from NANOZR.
- Unnecessary coloring of the graphs should be avoided, if colors do not have any specific meaning then b/w or grayscale should be preferred.
Thank you for your suggestion. We already use the colored graphs in another IJMS article. Each dissertation requires a graph in a color that is as easy to see as possible in the graduate student's degree application dissertation, so a colored graph is used.
- Terada C, Komasa S, Kusumoto T, Kawazoe T, Okazaki J. Effect of amelogenin coating of a nano-modified titanium surface on bioactivity. Int J Mol Sci 2018, 19(5).
- Nishizaki, M.; Komasa, S.; Taguchi, Y.; Nishizaki, H.; Okazaki, J. Bioactivity of NANOZR induced by alkali treatment. Int J Mol Sci 2017, 18, 780.
- Komasa, S.; Nishizaki, M.; Zhang, H.; Takao, S.; Kobayashi, Y.; Kusumoto, T.; Yoshimine, S.; Nishizki, H.; Okazaki, J.; Chen, L. Osseointegration of alkali-modified NANOZR implants: an in vivo study. Int J Mol Sci. 2019, 20, 842.
- The use of abbreviations/long forms is not uniform (such as it should either “day” or “d” not a mixture).
Thank you for your suggestion and have corrected the use of abbreviations/long form.
- Scale legends in Fig. 5 cannot be read.
Thank you for your suggestion. We changed scale legends in Fig. 5.
Reviewer 2 Report
Presented studies are focused on the determining the impact of plasma treatment on the properties of selected nanocomposites considering for application in implantology. Paper is valuable and provides interesting information, i.e. it was proved that the mentioned nanocomposite treatment resulted e.g. in the improvement of its biocompatibility. Therefore, the proposed article is worth considering for publication but some revisions are suggested – all of them are in more detail described below.
- Keyword “NANOZR” is not adequate, because it represents the abbreviation which is incomprehensible for readers unfamiliar with the article. It needs to be replaced by e.g. “Ce-stabilized zirconia/alumina nanocomposite”.
- Introduction: it was mentioned that various surface-roughening techniques have been investigated to enhance the biological activity and osteointegration capabilities of titanium. Some sample techniques should be listed.
- Section 2.1.: the discussion over the impact of plasma treatment on the contact angle of such treated materials should be extended.
- Captions of Figures should be concise and name the figure content briefly without providing any discussion or conclusions.
- 2.: the unit of protein adsorption should be added.
- Paper contains some grammar mistakes that should be corrected (e.g. line 147 - it should be “higher” instead of “high” etc.).
- Conclusions of the paper are too concise and should provide information concerning also the impact of plasma treatment on the contact angle or the surface morphology of the materials.
- Some references contain the whole journal names instead of their abbreviations (e.g. 49-53; 58-59; 61).
Author Response
Thank you very much for your comments. We have revised our manuscript in accordance with your suggestions as follows:
- Keyword “NANOZR” is not adequate, because it represents the abbreviation which is incomprehensible for readers unfamiliar with the article. It needs to be replaced by e.g. “Ce-stabilized zirconia/alumina nanocomposite”.
Thank you for your suggestion. We have corrected keywords from NANOZR to “Ce-stabilized zirconia/alumina nanocomposite”.
- Introduction: it was mentioned that various surface-roughening techniques have been investigated to enhance the biological activity and osteointegration capabilities of titanium. Some sample techniques should be listed.
We agree with your suggestion and have corrected our manuscript as follows:
Introduction:
Therefore, various surface-roughening techniques have been investigated to enhance the biological activity and osseointegration capabilities of titanium, including physical approaches such as compaction of nanoparticles [10], ion beam deposition [11]; chemical methods such as acid etching, peroxidation [12], anodization [13]; nanoparticle deposition such as discrete crystalline deposition [14] and lithography and contact printing technique [15]; however, little is known about the clinical effects of zirconia implants.
- Section 2.1.: the discussion over the impact of plasma treatment on the contact angle of such treated materials should be extended.
We agree with your suggestion and have corrected our manuscript as follows:
Page 8, lines 215-220:
When titanium or zirconia is subjected to plasma treatment, organic substances are decomposed by collision of high-energy molecules. In addition, the oxide is excited by oxygen in the plasma state, and at the same time, water molecules in the environment react to form a hydroxyl group. It is presumed that the carbon decomposition effect on the nano-ZR surface and the introduction of hydroxyl groups exerted superhydrophilicity.
- Captions of Figures should be concise and name the figure content briefly without providing any discussion or conclusions.
Thank you for your suggestion. We corrected figure legends.
- 2.: the unit of protein adsorption should be added.
Thank you for your suggestion. We added the unit of protein adsorption. (μg/dl)
- Paper contains some grammar mistakes that should be corrected (e.g. line 147 - it should be “higher” instead of “high” etc.).
We agree with your suggestion and have changed by English editing service (Editage). We also added the certificate of English editing.
- Conclusions of the paper are too concise and should provide information concerning also the impact of plasma treatment on the contact angle or the surface morphology of the materials.
We agree with your suggestion and have corrected our manuscript as follows:
Page 13, conclusion:
The results of the present study demonstrated that plasma treatment applied to alkali-modified nano-ZR surfaces increased the biocompatibility nano-ZR -based materials. The surface analysis of the concentrated alkali-treated nano-ZR treated with atmospheric pressure plasma showed no difference in the mechanical structure of the surface, but showed a decrease in carbon on the surface, indicating super hydrophilicity. The treatment enhanced the angiogenic induction and osteogenic induction abilities of alkali-modified nano-ZR, as well as the adherence of HUVECs and RBMCs. In addition, when the plasma-treated alkali-modified nano-ZR screw was implanted into a rat femur, compared to the untreated group, we showed that more new bone was formed in the tissue surrounding the implant. Therefore, based on both in vitro and in vivo results, we conclude that the atmospheric pressure plasma-treated nano-ZR material subjected to a concentrated alkali treatment and atmospheric pressure plasma treatment is useful as a prosthetic treatment option for patients allergic to metal.
- Some references contain the whole journal names instead of their abbreviations (e.g. 49-53; 58-59; 61).
Thank you for your suggestion. We corrected the journal names of (49-53, 58-59, 61).
Round 2
Reviewer 1 Report
The authors did not perform the bone mineral density measurements that are of critical importance for the paper, as indicated in the previous reviewer report. So, it is now, the editor´s decision to accept or reject this manuscript without bone mineral density measurements.